# Degradation bottlenecks and resource competition in transiently and stably engineered mammalian cells

Jacopo Gabrielli[1,2,3], Roberto Di Blasi [1,2,3], Cleo Kontoravdi [1,2] &
Francesca Ceroni [1,2] ✉

Degradation tags, otherwise known as degrons, are portable sequences that can be used to alter protein stability. Here, we report that degron-tagged proteins compete for cellular degradation resources in engineered mammalian cells leading to coupling of the degradation rates of otherwise independently expressed proteins when constitutively targeted human degrons are adopted. We show the effect of this competition to be dependent on the context of the degrons. By considering different proteins, degron position and cellular hosts, we highlight how the impact of the degron on both degradation strength and resource coupling changes, with identification of orthogonal combinations. By adopting inducible bacterial and plant degrons we also highlight how controlled uncoupling of synthetic construct degradation from the native machinery can be achieved. We then build a genomically integrated capacity monitor tagged with different degrons and confirm resource competition between genomic and transiently expressed DNA constructs. This work expands the characterisation of resource competition in engineered mammalian cells to protein degradation also including integrated systems, providing a framework for the optimisation of heterologous expression systems to advance applications in fundamental and applied biological research.

Protein degradation is a physiological process conserved across organisms that regulates cellular protein homeostasis[1–4]. In mammalian cells, degradation occurs primarily through the ubiquitin-proteasome system (UPS). This involves three subsequent enzymatic steps mediated by the ubiquitin-activating enzyme E1, the ubiquitin-conjugating enzyme E2, and ubiquitin-protein ligase E3, respectively, which facilitate protein poly-ubiquitination and subsequent proteasomal degradation. E1 mediates the binding of ubiquitin through an ATP-dependent reaction, subsequently transferring it to E2. E3 then catalyses the transfer of ubiquitin onto the target protein by formation of an isopeptide bond. By repeated rounds of ubiquitination (i.e. poly-ubiquitination) the target proteins are directed to degradation through the proteasomal pathway[1,5–7].

The likelihood of a protein being degraded by the UPS can be enhanced by the addition of a sequence of amino acids known as a degron. Degrons can be present within the protein sequence, at its C-terminus or at the N-terminus. C-terminal and N-terminal degrons have the advantage of being portable from one sequence to another, thus enabling controlled degradation of any protein of interest[8–11].

In synthetic biology, the number of proteins produced by engineered synthetic constructs and their degradation rate can drastically affect the function of gene and protein circuits. For this reason, degrons have gained popularity, with a variety of sequences developed as tools to achieve post-translational regulation of protein expression[12–15]. Among the applications, degrons have been used to accelerate transitions between gene circuit steady states[16–19] and to achieve desired dynamic expression patterns[20]. While some degron sequences mediate constitutive protein targeting by the degradation machinery (constitutive degrons), inducible degrons that enable

[1]Department of Chemical Engineering, Imperial College London, London, UK. [2]Imperial College Centre for Synthetic Biology, Imperial College London, London, UK. [3]These authors contributed equally: Jacopo Gabrielli, Roberto Di Blasi. ✉e-mail: f.ceroni@imperial.ac.uk

control of protein degradation upon addition of a specific inducer[15,21–26] have also been adopted, with applications in the regulation of chimeric antigen receptor activity and cell therapies[27–29].

One aspect that is usually overlooked in studies aimed at the design and characterisation of synthetic degradation systems is their impact on the cellular pool of degradation resources. In bacteria, Hasty et al. showed how using multiple degrons that share part of the degradation pathway, in their case the ClpXP protease, led to coupling of degradation rates and enzymatic queueing[30,31]. Mather et al. expanded the concept of competition for degradation resources in bacteria by characterising crosstalk involving other bacterial proteases such as ClpAP and Lon[32,33]. Previous work on resource competition in mammalian cells characterised bottlenecks in the transcriptional, translational and secretory machinery during transient gene expression[34–38]. However, considerations of how limitations in protein degradation resources impact heterologous gene expression have not been reported for these systems, yet[39].

Here we developed a framework for the characterisation of resource competition in mammalian cells at the protein degradation level, featuring both transient and stably integrated systems. We started by selecting a small number of degrons from a recent study by Chassin et al. that reported the comprehensive characterisation of a library of degrons in mammalian cells[17]. Three classes of degrons were considered. First, two classes featuring an N-terminal ubiquitin protein, whose N-terminal fusion is known to mediate proteasomal degradation, bearing either an intact or a mutated isopeptidase site. An intact isopeptidase site leads to recognition by deubiquitinating enzymes, which cleave ubiquitin, revealing either a stabilizing or destabilizing amino acid at the N-terminus of the target protein to which the N-end rule applies[10]. Conversely, degradation domains with a mutated isopeptidase site evade recognition by isopeptidase enzymes targeting the protein to the UFD (Ub-Fusion Degradation)[40]. Both categories of degradation domains lead to degradation via the proteasome-mediated pathway[17]. The third class included PEST (proline, glutamate, serine, threonine) degrons, posited to be independent of the ubiquitin proteasome system[41–44].

Our selected library included (i) two N-terminal ubiquitin tags featuring mutated isopeptidase sites, UbVR with a higher degradation rate, and 2 × UbAV with two mutated ubiquitin domains and a lower degradation rate, (ii) two N-terminal ubiquitin tags with intact isopeptidase sites, one with an N-terminal arginine and higher degradation rate, UbR, and one with an N-terminal methionine and a lower degradation rate, UbM, (iii) two C-terminal PEST-based degrons, MODCPEST with a higher degradation rate, and PEST with a lower degradation rate.

We used these degrons to generate a library of EGFP expressing test constructs encompassing C-terminal and N-terminal degradation aiming at characterising differences in resource availability between the C-terminal and N-terminal degradation pathways. We also widened our analysis to capture the effect that context exerts on competition for degradation. We considered the impact of protein context, replacing the EGFP with an mKate, and cell context, performing experiments in two different cell systems, HEK293T and CHO-K1. We then tested two widely adopted bacterial and plant degrons which provide the advantage of controlled degradation and are thus useful for synthetic biology applications where dynamic behaviours are sought. Subsequently, we constructed and characterised a genomically integrated, inducible, mammalian capacity monitor and employed this to expand considerations of resource competition to genomically expressed constructs. Our study provides characterisation of competition for degradation resources in mammalian cells and suggests routes to achieve uncoupling between transiently expressed systems and the cellular degradation machinery to improve the engineering of heterologous genetic systems with desired behaviour. The capacity monitor cell line developed here can find applicability as a proxy measure for impact of synthetic gene circuits on native gene expression and protein degradation.

## Results

### Competition for degradation resources impacts mammalian expression systems

Capacity monitors have been previously adopted to quantify and visualise gene expression resource competition in bacteria and mammalian cells[34,35,45–48]. Monitors are fluorescent expression cassettes that function as proxy for the available gene expression capacity within a cellular host. When adopted in combination with co-expressed test constructs, a change in the monitor's expression level can be considered a semi-quantitative proxy measure of the resource footprint of that given construct.

To investigate whether competition for degradation resources also impacts mammalian heterologous expression, we devised an experiment in which a PEST-tagged capacity monitor, designed by incorporating the PEST degradation domain into the fluorescent readout of transiently expressed monitor (mKate), was co-transfected either with a MODCPEST-tagged or an untagged EGFP-expressing test construct and performed a parallel control experiment where the mKate monitor is instead not tagged for degradation (Fig. 1A). All constructs shared identical plasmid backbones, promoters, Kozak sequences, and polyA signals. The only distinction between them is the presence or absence of the MODCPEST domain.

As previously observed for bacteria[31,33], our expectation was that competition for degradation resources between the MODCPEST-tagged test plasmid with the PEST-tagged capacity monitor would result in coupling of the intracellular levels of the two fluorescently tagged proteins indicating a possible bottleneck or queueing effect in the degradation pathway. This observation would be in line with the notion that the MODCPEST-tagged EGFP necessitates more protein degradation resources than its untagged counterpart, consequently reducing the availability of resources for efficient degradation of mKate-PEST at its usual rate and thus leading to increased mKate expression.

When MODCPEST was added to the EGFP-expressing construct, a reduction of the intracellular EGFP levels to 37% in HEK293T and 31% in CHO-K1 was observed compared to the untagged construct. This was mirrored by a 2.7- and 3.2-fold increase in mKate from the co-transfected capacity monitor (Fig. 1B). Such increase was not observed in the absence of PEST on the mKate coding sequence (Fig. 1C), suggesting the observed restoration was not attributed to increased expression levels of the mKate gene but rather a result of reduced degradation of mKate, a phenomenon akin to previously documented evidence in bacterial systems[31,33]. Our experiment indicates that the change in the capacity monitor's degradation is caused by the addition of a MODCPEST signal to the competing test construct, signifying a competition for resources. The MODCPEST-tagged EGFP variant requires more protein degradation resources compared to its untagged counterpart, thereby diminishing the resources available for efficient degradation of the PEST-tagged capacity monitor at its typical rates and leading to intracellular accumulation of the mKate protein.

### A library of degradation domains identifies specific degradation resource bottleneck

Once confirmed that we could adopt transient monitors to track competition for degradation resources in mammalian cells, we designed a suite of transient capacity monitors and test constructs that would enable us to assess and characterise the resource footprint of different degradation domains (Fig. 2A). We considered N-terminal and C-terminal degrons, both stronger and weaker than MODCPEST, as per their prior characterization[17]. We thus designed a library of seven EGFP expressing constructs bearing different degrons (UbVR, 2xUbAV, UbR, UbM, MODCPEST, PEST and a composite degron with an N-terminal UbVR and a C-terminal PEST).

In order to run competition assays, we designed two additional transient capacity monitors utilising UbR and UbVR. By co-transfecting different combinations of test constructs and capacity monitors and

**A)**

**B) mKate-MODCPEST Monitor**

Change in Capacity

| | |
|---|---|
| Untagged | 100% ± 16% |
| MODCPEST | 31% ± 4% |

Change in Capacity

| | |
|---|---|
| Untagged | 100% ± 34% |
| MODCPEST | 37% ± 17% |

**C) Untagged mKate Monitor**

Change in Capacity

| | |
|---|---|
| Untagged | 100% ± 22% |
| MODCPEST | 110% ± 21% |

Change in Capacity

| | |
|---|---|
| Untagged | 100% ± 47% |
| MODCPEST | 117% ± 27% |

assessing EGFP and mKate fluorescence by flow cytometry at 48 h post transfection, we expected to map competition patterns for different degrons and identify degron-specific competition. To capture the impact of the cellular background on competition, we performed the analysis in both CHO-K1 and HEK293T cells.

In CHO-K1 cells, the C-terminal degrons PEST and UbVR-PEST showed a higher propensity for coupling by causing an increase in the levels of the capacity monitor, irrespective of the capacity monitors tested, up to ~3.3-fold difference (Fig. 2B). In HEK293T cells, only MODCPEST induced significant coupling when co-transfected with the PEST monitor, leading to a ~2.7-fold increase in mKate (Fig. 2B). The PEST and MODCPEST degrons resulted in mild coupling with the UbR monitor both with a ~1.4-fold increase in mKate (Fig. 2B). In the case of the N-terminal degron UbR as capacity monitor, competition was

**Fig. 1 | Probing degradation-induced coupling via capacity monitors.**
**A** Simplified diagram of the UPS degradation pathway followed by experimental workflow schematics: a library of test constructs, tagged with degrons, is co-transfected into HEK293T and CHO-K1 cell lines along with a destabilized capacity monitor. Intracellular protein levels are quantified using flow cytometry. The competition for degradation resources can be visualized by plotting the levels of the capacity monitor protein against the levels of the test constructs' protein. In this plot, the constructs located at the bottom left corner represent the most efficient candidates. **B** Proof-of-concept experiment where a destabilized monitor, with a PEST degron, is co-transfected with a test construct tagged with the MODCPEST degron in HEK293T and CHO-K1 cells. **C** Proof-of-concept experiment where an

untagged stable monitor is co-transfected with a test construct tagged with the MODCPEST degron in HEK293T and CHO-K1 cells. Test plasmid and capacity monitor intracellular protein levels are reported as mean fluorescence (arbitrary units) ± std. The table reports 1/(mKate of EGFP-tagged condition/mKate of EGFP untagged condition) to quantify change in degradation capacity upon adding the degron ± the ratio of the standard deviation to the mKate mean multiplied by change in capacity. Data derive from three independent experiments each comprised of two biological repeats. A two-tailed Student's $T$-Test was used, where $P$ values are denoted as follows: *****<0.00005, ****<0.0005, ***<0.0005, **<0.005, *<0.05. The number of biological repeats for each sample and exact $P$ values are reported in Source data file.

observed against UbVR with a ~1.9-fold increase, 2xUbAV with ~2.2-fold and UbR with ~2.7-fold, while with the UbVR monitor the same were ranked highest but with a much milder effect: ~1.6-fold, ~2-fold and ~1.1-fold increase in mKate, respectively (Fig. 2B). Interestingly, competition for degradation resources in our experiments did not appear to be linearly correlated with the strength of the degrons employed. As a control, the same experiment was performed in parallel with EGFP constructs tagged with the different degrons co-transfected with untagged mKate (Fig. S1). Again, we confirmed that untagged mKate monitor levels do not increase above the levels displayed by mKate in competition with an untagged EGFP with statistical significance for any of the combinations. We speculate that the increase in monitor expression observed when the monitor is tagged for degradation (Fig. 2) is due to competition for degradation. Interestingly, we observed that for some of the degrons, untagged mKate expression decreases below the levels observed when mKate competes with untagged EGFP, suggesting that, for some combinations, the presence of the degron leads to increased competition for gene expression upstream of degradation possibly due to impact in transcription and translation efficiency.

To capture not only the impact of the cell host but also the one of the protein context on competition, we then assembled a new library of constructs where mKate bears the same degrons previously used for EGFP and EGFP now functions as the capacity monitor. Experiments were performed in both HEK293T (Fig. S2) and CHO-K1 cells (Fig. S3). For HEK293T cells, competition was only detected with UbR and UbM N-terminal degrons, while in CHO-K1 cells, no significant increase in monitor levels was detected. We also performed experiments with their respective untagged EGFP monitor controls, similar to what previously done for the mKate monitor (Fig. S4). Again, we did not observe any statistically significant increase in the monitor level, as previously confirmed for the untagged mKate control. Notably, for PEST, MODCPEST and UbVR-PEST in CHO-K1 an increase in the level of mKate construct expression was observed (Fig. S4), potentially providing an explanation for the competition pattern observed for the EGFP-monitors in Fig. S3. Overall, from our data, protein context, cell host, and, potentially, position of the degron on the protein sequence, appear to play a major role in shaping competition for protein degradation.

### Testing of inducible degradation domains
Inducible degrons can be used to trigger protein degradation upon addition of a chemical. Their fast dynamics and ease of control through cell-permeable small molecules have led to inducible degrons being adopted as tuning dials for protein regulation both in synthetic biology and cell therapy[21–23,27].

Here we tested the effect of two commonly used inducible degradation domains, mAID2, an auxin-inducible degron[23] and ecDHFR-DD, a trimethoprim (TMP) inducible degron[22], on a co-transfected UbVR-tagged capacity monitor. Both domains were fused to the N-terminus of EGFP to resemble UbVR, also located at the N-terminus. While both systems allow for inducible control over the stability of a protein of interest, they do have significant differences.

The *Escherichia coli* dihydrofolate reductase protein (ecDHFR) is a 45-kDa degradation-prone domain stabilized through the introduction of the small molecule antibiotic TMP[22] (Fig. 3A). We cloned a plasmid containing a DHFR-EGFP fusion protein and co-transfected it alongside our UbVR-mKate capacity monitor design.

Addition of 0.1 μM of TMP, the concentration which was previously shown to maximise stabilization in mammalian cells[22], did not lead to a statistically significant change in intracellular mKate levels, while intracellular EGFP levels increased up to ~14-fold (Fig. 3B). The increase in EGFP suggests the correct functioning of the N-terminal DHFR domain fused to EGFP in both cell lines. Therefore, the stability of mKate levels across conditions concomitant with a reduction in EGFP degradation in both cell lines suggest that degradation resources required for the TMP-inducible degron are at least partially orthogonal to the ones required for the N-terminal degron UbVR, similarly to previous work[17].

The auxin-degradation technology relies on three key components: the *Oryza Sativa* TIR1 protein F74G (OsTIR1 F74G), a 7-kDa degron known as mAID2 fused to the protein of interest, and the small-molecule auxin. When expressed in mammalian cells, OsTIR1 forms an Skp1-Cul1-Fbox (SCF) E3 ligase complex with endogenous components. In the presence of 5-Phenyl-1H-indole-3-acetic acid (5-Ph-IAA), this complex binds to mAID2, leading to the ubiquitination and subsequent proteasomal degradation of the protein to which it is fused (Fig. 3C). This technology enables efficient protein degradation in human cells expressing OsTIR1, with a half-life of 60–65 minutes[23].

When co-transfecting the mAID2-tagged EGFP with the UbVR-tagged capacity monitor, a decrease between 3.5 and 20-fold in EGFP levels upon the addition of 0.1 μM of the inducer molecule 5-Ph-IAA was observed across cell lines, while we failed to observe significant differences in mKate (Fig. 3D). Similarly to our TMP-inducible degron, these findings suggest that degradation resources required for the 5-Ph-IAA-inducible degron are at least partially orthogonal to the ones required for the N-terminal degron UbVR.

### Competition for gene expression and degradation resources affects genomically integrated genes
To investigate whether competition for degradation resources impacts the behaviour of integrated synthetic cassettes, we constructed a capacity monitor cell line in HEK293T. To achieve fast, sensitive and controllable response to resource uptake from transcription to degradation, we designed the monitor to be inducible and to bear N- and C-terminal degradation tags.

We cloned a construct where, upon doxycycline induction, the TET promoter (TETp)[49] drives the expression of three mCherry genes fused together and appended with an N-terminal and C-terminal degradation domains, UbVR and PEST, respectively. We integrated this design at the AAVS1 locus in HEK293T cells with a CRISPR-Cas9 all-in-one plasmid[50] (Fig. 4A) (see also the Methods section and Supplementary Note 2). We selected one clone for further experiments as it displayed correct PCR-genotyping profile and the expected induction behaviour in the presence of the inducer (Fig. 4B).

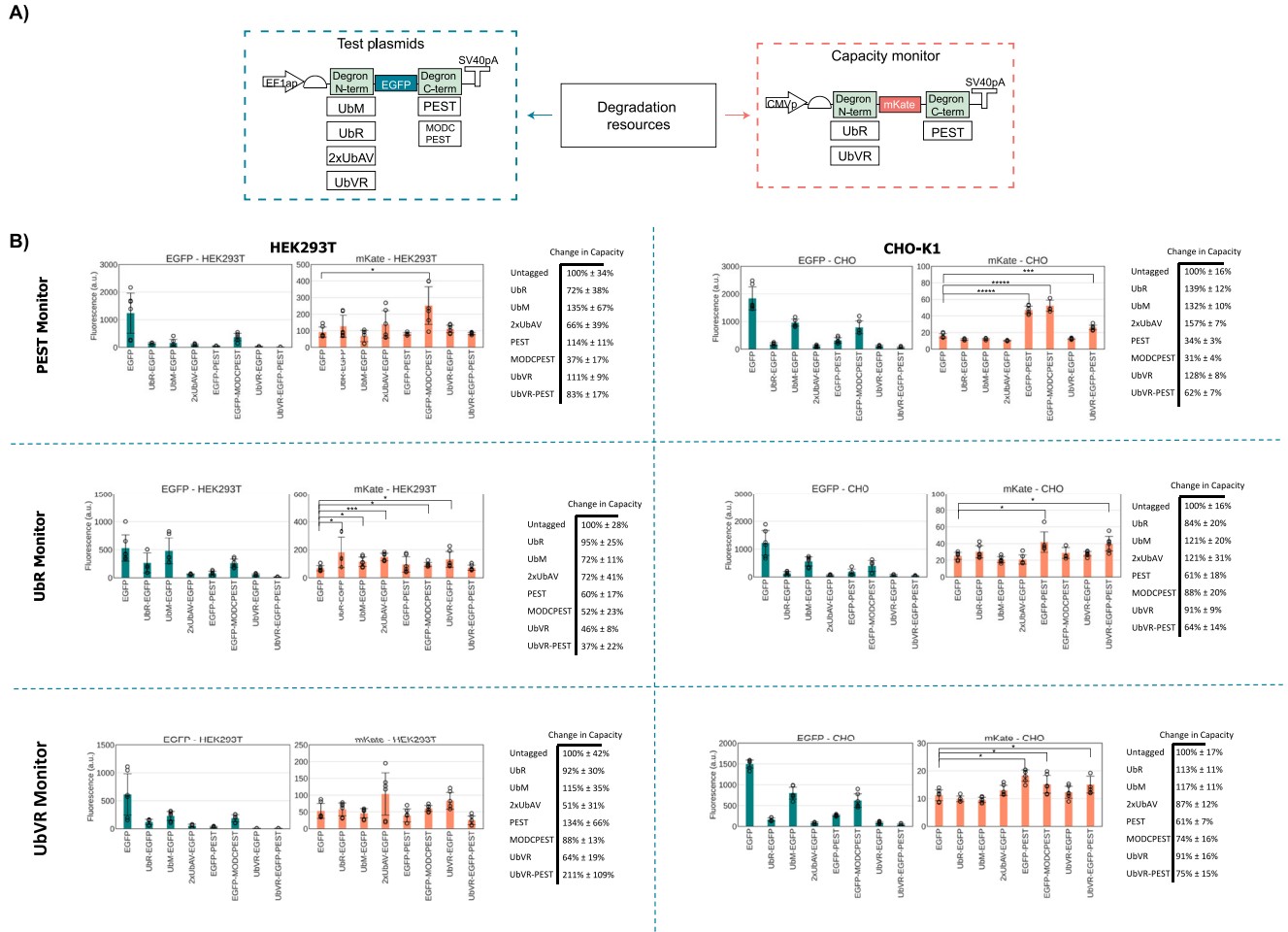

**Fig. 2 | Characterizing a library of degrons with three co-transfected capacity monitors. A** Schematic of the genetic construct library considered. Three degradation monitors tagged with UbR, UbVR and PEST respectively were co-transfected with seven test constructs tagged with UbM, UbR, 2xUbAV, UbVR, PEST or MODCPEST. **B** Flow cytometry data for monitor with PEST (top), UbR (middle) and UbVR (bottom), in HEK293T (left) and CHO-K1 cells (right). Test plasmid and capacity monitor reporter levels are reported as mean fluorescence (arbitrary units) ± std. Data derived from three independent experiments each with two biological repeats. A two-tailed Student's $T$-Test was used, where $P$ values are denoted as follows: ***** < 0.00005, **** < 0.0005, *** < 0.0005, ** < 0.005, * < 0.05, and it was annotated only on the mKate charts with reference to the control, EGFP versus capacity monitor, for clarity purposes. Tables report 1/(mKate of EGFP-tagged condition/mKate of EGFP untagged condition) to quantify change in degradation capacity upon adding the degron ± the ratio of the standard deviation to the mKate mean multiplied by change in capacity. The number of biological repeats for each sample and exact $P$ values are reported in the Source Data File.

Since competition for transcriptional and translational resources by capacity monitor measures in mammalian cells has only been tested in transient systems[34–36], we first investigated if an integrated monitor can quantify competition for gene expression resources.

Previous work highlighted that transcriptional resource load plays a major role in determining the footprint of a genetic constructs on the host gene expression resources[35,36]. We started by considering monitor expression when the monitor cells are transiently transfected with a library of constructs that only differ in their EGFP promoter (Fig. 4C).

The library was previously tested in Di Blasi et al. [35] and includes seven constitutive promoters, the two viral CMVp (Cytomegalovirus promoter) and SV40p (Simian Virus 40 promoter), and five human, pJB42CAT5 (a minimal promoter derived from JunB) hACTBp (human β-actin), EF1αp (elongation factor 1α promoter), UBp (Ubiquitin promoter) and the PGKp (3-phosphoglycerate kinase promoter).

Increasing promoter strength on the EGFP constructs yielded up to ~11-fold decrease in monitor expression, confirming that the integrated monitor is sensitive to increasing transcriptional resource uptake. A single exception was represented by hACTBp, resulting in reduced monitor expression capacity by ~56% but yielding EGFP levels

considerably higher than more burdensome promoters (Fig. 4D). This is in line with previous work highlighting that some promoter pairs can lead to a trend different from the linear decrease in monitor expression with higher EGFP expression usually expected[35]. Growth was shown to linearly correlate with monitor expression, with decreasing cell counts observed for decreasing levels of the capacity monitor, mKate expression (Fig. S5).

Once confirmed that the monitor can be adopted as proxy for competition between genomically expressed and transient genetic cassettes, we extended our analysis to degradation resources. We transfected the HEK293T capacity monitor cell line with test plasmids encoding either an untagged EGFP or an EGFP fused with UbVR, PEST or both UbVR and PEST degradation domains (Fig. 4E). Remarkably, upon transfection of PEST and UbVR-PEST tagged constructs, we observed increased accumulation levels of the capacity monitor, 1.49-fold ($p$-value = 0.0002) and 1.33-fold ($p$-value = 0.005), respectively (Fig. 4F). Interestingly, while the UbVR-tagged EGFP experienced more degradation than the PEST-tagged EGFP, 9.71-fold reduction ($p$-value = 0.0005) versus 3.38 ($p$-value = 0.005) compared to the untagged construct, the latter resulted in higher accumulation levels of the integrated capacity monitor, 1.49-fold ($p$-value = 0.0002) increase versus 1.22-fold

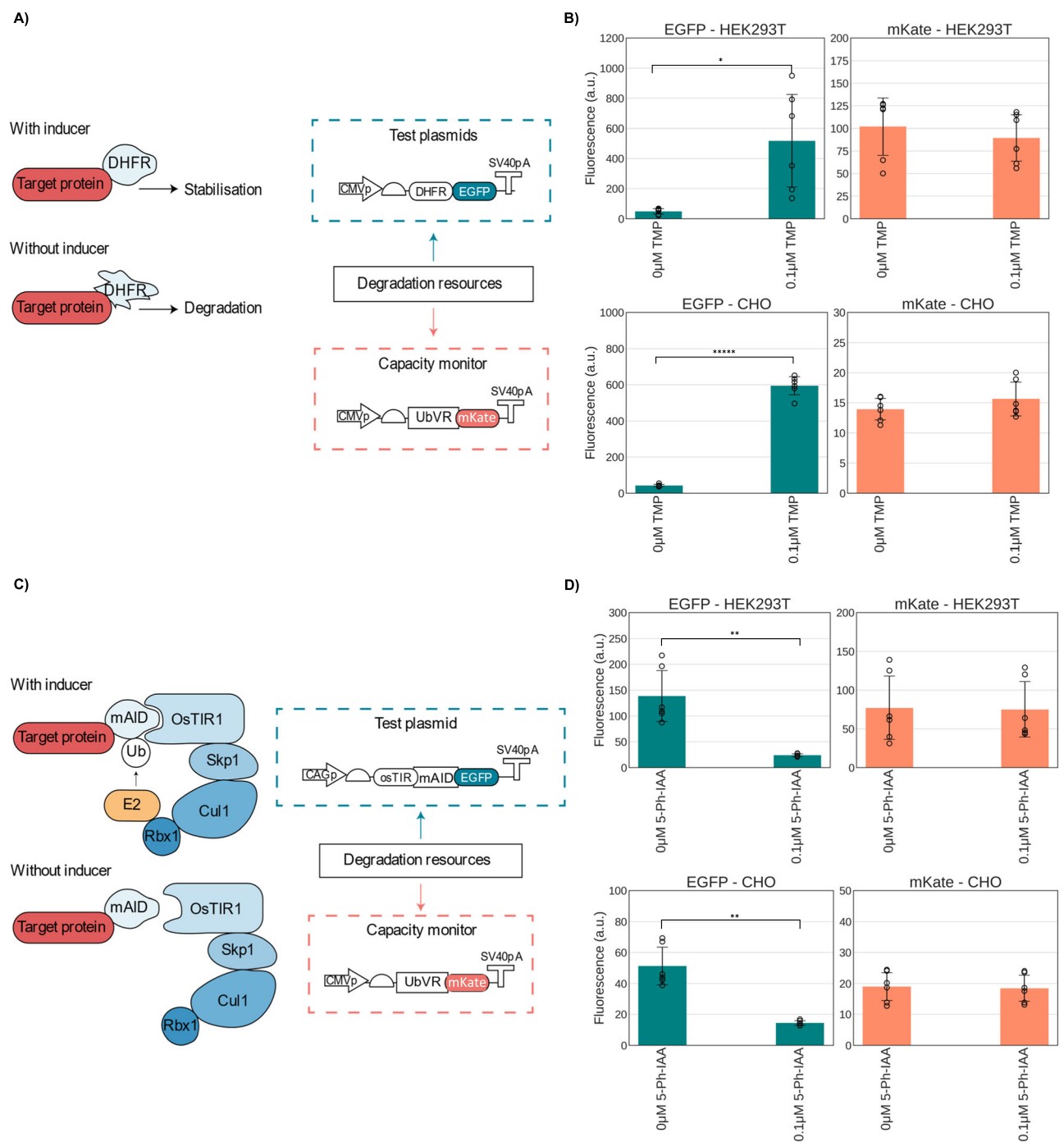

**Fig. 3 | Testing bacterial and plant degrons for coupling in mammalian cells.**
**A** Schematic diagram of the TMP-inducible DHFR degron and the experimental setup. The degron is inherently destabilizing, TMP binds it and stabilizes it. These were tested against the UbVR degron and their uninduced condition was compared to the induced one. **B** The UbVR destabilized monitor was co-transfected in both CHO-K1 and HEK293T with an EGFP bearing the TMP degron either uninduced or induced with 0.1 μM of TMP. **C** Diagram of the 5-Ph-IAA-inducible mAID2 degron and experimental setup. OsTIR1 forms an Skp1-Cul1-Fbox (SCF) E3 ligase complex with endogenous mammalian components. The degron is activated upon addition of 5-Ph-IAA which coordinates binding of the OsTIR1 E3 ligase to the mAID2 degron. These were tested against the UbVR degron and their induced condition was compared to the uninduced one. **D** The UbVR destabilized monitor was co-transfected in both CHO-K1 and HEK293T with a construct encoding the OsTIR1 E3 ligase and EGFP bearing the mAID2 degron either uninduced or induced with 0.1 μM of 5-Ph-IAA. Test plasmid and capacity monitor intracellular protein levels are reported as mean fluorescence (arbitrary units) ± std. Data derive from three independent experiments each comprised of two biological repeats. A two-tailed Student's *T*-Test, where *P* values are denoted as follows: *****<0.00005, ****<0.0005, ***<0.0005, **<0.005, *<0.05. The number of biological repeats for each sample and exact *P* values are reported in the Source Data File.

(*p*-value = 0.01) decrease (Fig. 4F). This observation aligns with our earlier findings regarding the competition of two C-terminal degradation tagged proteins (Fig. 1B, Fig. 2B). It further suggests that certain critical C-terminal or PEST-specific cellular resources may become limited when two PEST-tagged proteins are expressed simultaneously.

## Discussion

Degrons are amino acid sequences that modulate the degradation rate of the protein they are fused to, often by triggering the ubiquitin-proteasome system. They have been widely adopted in synthetic biology to tune protein degradation rate and achieve desired

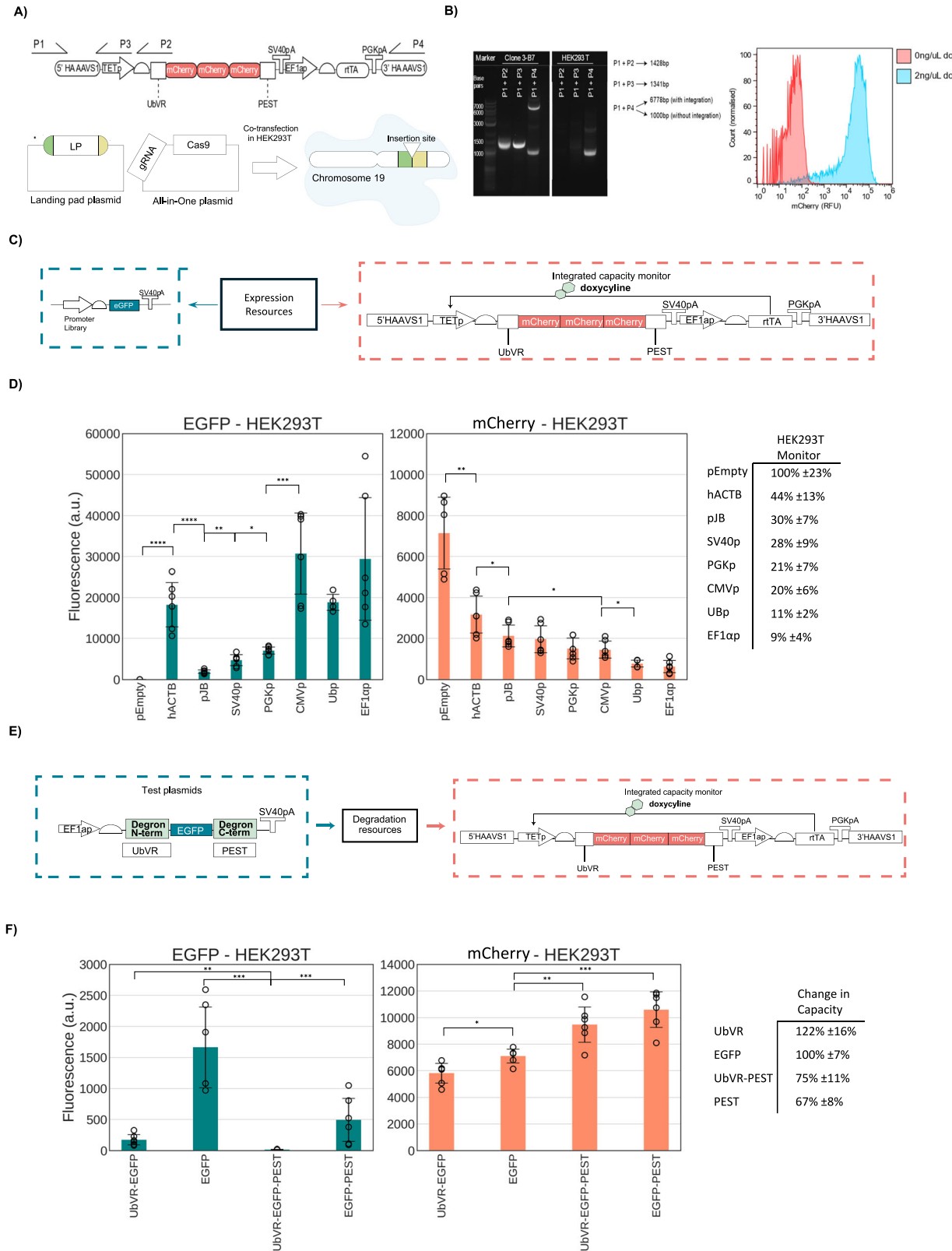

functionalities[8,13,17,21,27,28]. In bacteria, previous research highlighted how degradation of engineered heterologous proteins can suffer from bottlenecks derived from limitations in the cellular protein degradation machinery[30–33]. As of today, such a characterisation is still lacking for mammalian cells limiting our ability to include considerations of such limitations at the design stage and improve predictability.

Here we characterise the resource footprint of a diverse library of N-terminal and C-terminal degradation domains when degron-bearing constructs are co-transfected with a transient capacity monitor. In order to capture the effect of the cellular context on competition for degradation, we first considered the case of an mKate monitor bearing one of three degradation tags (PEST, UbR or UbvR) in competition with EGFP constructs bearing one of seven degradation tags (UbvR, 2xUbAV, UbR, UbM, MODCPEST, PEST, and a composite degron with an N-terminal UbvR and a C-terminal PEST) in two different cell lines, HEK293T and CHO-K1. Our data highlight that competition for degradation is cell line

**Fig. 4 | Resource competition and degradation coupling between transfected and integrated genes. A** Diagram of the construction of an integrated, degraded, capacity monitor in HEK293T. The monitor is composed of three mCherry repeats flagged by both N-terminal and C-terminal degrons. It was integrated with an All-in-one CRISPR system into chromosome 19 at the AAVS1 safe harbour locus. **B** Integration validation via PCR and flow cytometry of the expanded PCR-verified clone. The gel was performed once after 26 days from integration and the flow cytometry spectrum is representative of >3 independent repeats. **C** A library of test plasmids bearing different promoters was selected to test the monitor response to competition. **D** EGFP construct expression and mCherry monitor expression when the promoter library is adopted. A summary table reports (mCherry of EGFP expressing condition/mCherry of pEmpty condition). **E** Competition for

degradation resources with the integrated capacity monitor tested against EGFP bearing either UbVR, PEST or both. **F** Test plasmids were transfected along empty plasmids to match the total DNA transfected to the previous experiments. The table reports 1/(mCherry of EGFP-tagged condition/mCherry of EGFP untagged condition). Test plasmid and capacity monitor intracellular protein levels are reported as mean fluorescence (arbitrary units) ± std. In D data points lower than Q1-1.5*IQR and higher than Q3-1.5*IQR-where IQR stand for interquartile range-where considered outliers and excluded from further analysis. Data derive from three independent experiments each comprised. A two-tailed Student's *T*-Test was used, where *P* values are denoted as follows: *****<0.00005, ****<0.0005, ***<0.0005, **<0.005, *<0.05. The number of biological repeats for each sample and exact *P* values are reported in the Source Data File.

dependent. Specifically, in HEK293T, the UbVR-mKate monitor displayed no significant competition with any of the EGFP tagged constructs while the PEST-mKate monitor did not display competition with the corresponding PEST-tagged EGFP construct. This was not the case for experiments performed in CHO-K1 where we instead observed competition for the same combinations resulting in an increase in monitor expression levels. While we do not have a justification for the observed difference, we speculate that, as previously observed for competition imposed by other genetic elements like promoters and polyA signals[35], different cellular environments and differences in basal levels of resources related to gene expression regulation could lead to variations in competition profiles. Additionally, minor cell line specificity when comparing degron activity was also previously reported with multiple degrons including UbM and UbR also featured in our study, in HEK293, HeLa, hMSCtert and CHO cells[17]. Finally, the observed low levels of expression resulting from enhanced degradation could make it more challenging to detect a statistically significant resource competition effect given the intrinsic expression variability, which in our data appears higher in HEK293T than in CHO-K1.

To account for protein context effects on competition, we then performed experiments where the monitor is EGFP instead of mKate, adopting the same combination of degron tags. For mKate capacity monitors, C-terminal degrons such as PEST and MODCPEST showed a higher degree of coupling across cell lines, while when an EGFP capacity monitor is adopted, competition was only detected with UbR and UbM N-terminal degrons in HEK293T for two of the three monitors considered. Interestingly, in these experiments the UbVR-EGFP monitor displayed competition when co-transfected with the mKate library in HEK293T while the UbR-EGFP monitor did not yield significant competition, as previously observed for the UbR-mKate monitor. Previous studies reported how the degradation strength of a given degron is impacted by the protein associated with it, leading to significantly different degradation strength for the same degrons[15]. Degron-mediated degradation efficiency has also been reported to be dictated by the presence of specific structural and conformational elements. Missense mutations in human DHFR were found to cause conformational changes that led the wildtype DHFR to expose an otherwise hidden degron, with consequent increase in DHFR degradation[51]. Protein domains termed "degronons", while not directly recruiting the E3 ligase, were found to coordinate 26S proteasome binding its substrate. Therefore, changing the protein but maintaining the degron could weaken or strengthen its activity based on the presence or absence of degronons on the new protein[52].

In parallel, we also included control competition experiments between the same tagged EGFP/mKate constructs and untagged mKate/EGFP monitors to confirm that the increase in construct expression levels is indeed due only to competition for degradation. Our results confirmed that for untagged mKate and EGFP no significant increase in monitor levels occur as a consequence of competition. Interestingly, we also observed that different degrons led to more or less pronounced decrease of mKate/EGFP levels, suggesting that the presence of the degron impacts gene expression competition

upstream of degradation. This highlights a more complex scenario than previously hypothesised and adds to the importance to characterise the impact of context on the specific system considered.

We then quantified competition imposed by N-terminal inducible degrons, DHFR degradation domain and mAID2, respectively, and observed no significant coupling with the capacity monitor. We speculate that mAID2 may rely on partially orthogonal, plant-derived, E3 ligase complex, the Skp1-Cul1-OsTIR1[23,53]. While ecDHFR is a bacterial unstable domain, its degradation pathway in mammalian cells has not been fully elucidated at the molecular level[22]. Overall, our experiments highlight how systems like these inducible degrons could be adopted in applications where minimised competition and enhanced control of degradation are sought.

To investigate the impact of transiently expressed cassettes on genomically integrated genes, we then designed a capacity monitor in HEK293T that we first test for expression competition with a promoter library previously tested with transient monitors. The promoter library confirmed that competition for transcriptional resources, can be quantified by means of an integrated monitor showing an inverse linear correlation between expression levels of the transient construct and the integrated monitor with hACTBp as the only exception.

Moreover, the integrated capacity monitor, when tested with plasmids from our constitutive degrons library, confirmed coupling of intracellular resources between transient and integrated genes in the degradation pathway too. Similarly to the transient mKate monitor system, in the integrated monitor, C-terminal or PEST-like degrons appear to compete for a smaller pool of resources even though they exhibit degradation levels comparable to N-terminal degrons. Future fundamental research should focus on understanding the cellular repercussions of competition for degradation, as we show it can affect genomically integrated genes and targeted degradation therapeutics are entering the clinic.

The integrated capacity monitor presented here functions as a proxy measure of expression resource availability in HEK293T and provides an example of how integrated monitors could be leveraged in mammalian cell engineering. Our integrated monitor, while it is not specific like qPCR or RNA-seq in pinpointing changes in expression levels, is a proxy measurement of expression and degradation competition and its impact on the cell, offering a complementary approach to cell growth for burden assessment. Further research will be needed to elucidate the effect of this competition on a broader range of genomic genes and across different cell lines[54].

Overall, in the future, bottlenecks in protein degradation could be leveraged in synthetic biology to improve context-aware design of gene circuits exploiting degrons or to induce competition for degradation as a mean of synthetic regulation. For therapeutic purposes, competition for degradation could be used to control specific protein degradation pathways and to stabilise cellular proteins whose dysregulated degradation contributes to human diseases[55–57] surpassing the toxicity challenges of current proteasomal inhibitors[58–60] or to optimise the multiplexing of PROTACs or molecular glues because of their reliance on the endogenous degradation pathways[61–63].

Our data highlight how context dependencies, specifically the protein and cell host context together with position of the degron on the tagged protein, play a crucial role in shaping competition profiles and the activity of multiple mammalian degrons. They also identify designs with improved orthogonality in resource usage, making them amenable to synthetic systems with multiple degron-tagged proteins. Our findings thus underscore the importance of taking a more holistic approach when considering the impact of competition for gene expression resources in mammalian cells, including consideration for additional regulatory pathways such as protein degradation and secretion resources. Such a comprehensive approach is critical for unravelling the intricacies of cellular regulation when disrupted by expression of heterologous genes and could have broad-reaching implications for the fields of biotechnology and synthetic biology.

## Methods

### Plasmid design and preparation

The plasmids adopted in this study were built by PCR-based cloning. Starting from donor plasmids and a template plasmid, we added the degradation tags at the C and N terminus of the fluorescent protein. Fragments were isolated by gel electrophoresis (Agarose, Sigma-Aldrich, Gel Loading Dye, Purple(6x), NEB) and purified using a Qiagen extraction kit as per the manufacturer's instructions (QIAquick Gel Extraction Kit, Qiagen). A complete list of plasmids included in this study can be found in Supplementary Note 3. Plasmid maps annotated for specific parts and sequences used in this paper can be found in the Source data file. Primers used in this study can be found in Supplementary Data 1.

Templates and inserts were restricted and ligated with the EMMA protocol[64] using Esp3I (Esp3I, Thermo Scientific) and T4 DNA ligase (T4 Ligase, NEB). Ligations were transformed in chemically competent *E. coli* DH5α by heat shock at 42 °C for 30 s and 45 min outgrowth in 1 mL Luria Broth (LB Broth, Sigma-Aldrich). 100 μL were then plated on agar plates supplemented with 100 μg/mL ampicillin (Ampicillin, Sigma-Aldrich). The antibiotic-resistant colonies were then inoculated in 5 mL of Luria Broth the following day and incubated overnight at 37 °C. The culture was then midi prepped (CompactPrep Plasmid Midi Kit, Qiagen) according to the manufacturer's protocol the following day to extract the plasmid DNA, ready to be diluted for transfection.

### Mammalian cell culture and transfection

HEK293T cells (HEK293T, ATCC CRL3216) were cultured in T75 flasks (Nunc™ EasYFlask™ Cell Culture Flasks, Thermo Scientific) using DMEM (Dulbecco's modified Eagle Medium) supplemented with 1 mM sodium pyruvate, 1x GlutaMax™, 0.4 mM phenol red, and 25 mM glucose (DMEM, high glucose, GlutaMAX™ Supplement, pyruvate, Gibco), along with 10% FBS (Fetal Bovine Serum, Gibco). Cells were kept in an incubator at 37 °C and 5% CO₂ and were passaged upon reaching 70% confluency. 1 mL of trypsin-EDTA (0.5%, no phenol red, Gibco) was used for cell detachment during passages, and 1 mL of PBS (phosphate buffered saline, Merck) was used for cell washing.

CHO-K1s were cultured in MEMα (Minimum Essential Medium Eagle, Sigma) supplemented with 10% FBS (Fetal Bovine Serum, Gibco), 1% non-essential amino acids (MEM Non-Essential Amino Acids Solution (100X), ThermoFisher), and 1% L-glutamine (L-Glutamine (200 mM), ThermoFisher) in T75 flasks (Nunc™ EasYFlask™ Cell Culture Flasks, Thermo Scientific).

HEK293T cells were seeded 1 day before transfection at 100,000 cells/well in 24-well plates (Multiwell cell culture plates, VWR) for the transient transfections. Details on the transfection mixes can be found in Supplementary Data 2. The complete mix was incubated for 30 min pre-transfection and 2 days after.

CHO-K1 cells were seeded for transfection at 8 * 10⁴ cells/well in 24 w plates one day before transfection. Details on the transfection mixes can be found in Supplementary Data 2. The complete mix was incubated for 25 min at room temperature pre-transfection and 2 days after.

2 ng/μL of doxycycline (doxycycline, Sigma Aldrich) was added to the wells containing cells transfected with plasmids with a dox-inducible promoter for HEK293Ts on day 1. For both HEK293T and CHO-K1 0.1 μM 5-Ph-IAA (5-phenylalanine-indole acetic-acid) (5-Ph-IAA, Cambridge Bioscience) to the ones with an mAID2 tag and 0.1 μM TMP (Trimethoprim Ready Made Solution, Sigma Aldrich) to the ones with an ecDHFR tag right after transfection or at several time points.

### Stable cell line development

This study's CRISPR integrations involved integrative payloads expressing a fluorescence gene. This facilitated the selection of the integrated population and allowed easy assessment of the stability of the selected clones. All the integrative payloads in this study have been integrated into AAVS1 in HEK293T (described in Supplementary Note 2), using the following gRNA sequence, which was already experimentally validated[50]:
5′-GGGGCCACTAGGGACAGGAT-3′.

For CRISPR-based integrations, 2 * 10⁵ HEK293T cells were seeded in 24-well plates one day before transfection. The day after, 0.06 pmol of donor backbone and 0.04 pmol of All-in-One backbone were co-transfected in HEK293T using FuGENEHD (FuGENE® HD Transfection Reagent, Promega) as transfection reagent at a ratio of 3:1 FuGENEHD (μL): transfected DNA (μg). 24 h after transfection, the spent media was changed with fresh culture media and cells were cultured for 10 days at least. Cells were detached, and the positive fluorescent population for our marker was single-cell sorted into individual wells of a 96-well plate.

The resulting single-cell clones were cultured for three-to-four weeks in conditioned medium (i.e., filtered spent media collected from a confluent flask) at the ratio of 1:3 to fresh medium with 25% FBS and 1% penicillin-streptomycin (Penicillin-Streptomycin, Sigma Aldrich). Confluent wells were then assessed with fluorescence microscopes, and the ones displaying post-integration homogeneous fluorescence were sorted and expanded.

For cell sorting experiments during cell line development, cells were detached, centrifuged for 5 min at 200 × g, washed in DPBS, centrifuged again for 5 min, and resuspended in DPBS at a final concentration of 10⁷ cells/mL. Cells were kept on ice and filtered in test tubes with cell strainer caps to disrupt cell clumps. Sorting was carried out at 4 °C using a BD FACSAria Fusion (BD FACSAria Fusion, BD Biosciences) by specialized facility staff. Purity checks on the sorted population were carried out whenever possible.

As soon as they reached confluency in a 24-well plate format, the genomic DNA of the expanded clones was extracted with a kit according to the manufacturer's protocol (Nucleospin Tissue, Macherey-Nagel) to perform PCR genotyping. Correct clones were expanded further, aliquoted, and frozen in culture media with 10% DMSO (Dimethyl Sulfoxide, Merck).

### Detection of intracellular fluorescence via flow cytometry

At 48 h post-transfection, cells were detached by washing with 400 μL/well of PBS. The cell suspension was transferred to microcentrifuge tubes and centrifuged at 200 × g for 5 min. The resulting cell pellet was resuspended in 300 μL of PBS and transferred into test tubes with cell strainer caps using a filter (Corning™ Falcon™ Round-Bottom Polystyrene Test Tubes with Cell Strainer Snap Cap, 5 mL, FisherScientific). For analysis, 50 μL of each sample was loaded onto an Attune NxT (Attune NxT Flow Cytometer, Thermofisher) to collect approximately 10,000 single cell events. The EGFP fluorescence was excited with a 488 nm laser and detected using a 530/30 nm bandpass filter, the mKate fluorescence was excited with a 561 nm laser and detected using a 620/15 nm bandpass filter.

Flow cytometry data analysis was performed using the FlowJo software (FlowJo™, BD Life Sciences). Live cells were gated based on the FSC/SSC dot plots to exclude debris with low side and forward scatter. Single cells were gated based on the FSC-H/FSC-A plot. Then, the levels of EGFP and mKate were determined by calculating the

geometric mean of the entire population. Compensation of the red and green signals was automatically performed in FlowJo using a green control expressing only EGFP and a red control expressing mKate. The analysis and visualisation of processed EGFP and mKate values was carried out with both Microsoft Excel and Python 3.12 with the matplotlib package. Further details on the gating strategy can be found in Supplementary Note 1 and Fig. S6. Further detail on the processing of flow cytometry data can be found in the Source Data File.

**Live cell counting assay**

For live cell counting experiments, transfected cells in 24 well plates were washed in DPBS detached in 100 μL of 1% trypsin-EDTA, and 200 μL of DMEM + 10% FBS was added to a total volume of 300 μL. Lastly, 2.5 μL of solution 18 (Solution 18, Chemometec) was added to 50 μL of cell suspension and live cells were counted using the Nucleocounter (Nucleocounter NC250, Chemometec). Cell counts can be found in the Source Data File.

**Reporting summary**

Further information on research design is available in the Nature Portfolio Reporting Summary linked to this article.

## Data availability

All data generated in this study are provided in the Source Data and Supplementary Information files. Source data are provided with this paper.

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

## Acknowledgements

The authors would like to thank Tom Ellis, Mustafa Khammash and Velia Siciliano for early discussion on the project; Martin Fussenegger for sharing plasmid pCHX116 (2xUbAV), pCHX204 (UbM), pCHX247 (UbR) and pCHX155 (MODCPEST) with the degrons adopted in this paper; Ervin Welker for sharing plasmid pSC1-DD (EGFP-ecDFHR); Karen Polizzi for constructive feedback on the manuscript. The authors acknowledge the support of the EPSRC Centre for Doctoral Training in BioDesign Engineering (EP/S022856/1) (to J.G., C.K. and F.C.). R.D.B. was supported by the Imperial College Chemical Engineering PhD scholarship.

## Author contributions

All authors conceived the research and designed the experiments. JG and RDB performed the experiments. All authors wrote and edited the manuscript.

## Competing interests

The authors declare no competing interests.
