## [Transparent Peer Review file · Nature Communications]

Degradation Bottlenecks and Resource Competition in Transiently and Stably Engineered Mammalian Cells

Corresponding Author: Dr Francesca Ceroni

Version 0:

Reviewer comments:

Reviewer #1

(Remarks to the Author)

In this work, the authors investigated whether competition for cellular degradation resources could affect synthetic gene expression levels in a mammalian system. They utilized a library of degradation domains to characterize this competition in both transient and genomically integrated systems. Through systematic measurements across multiple cell lines and rigorous analysis, the authors confirmed that resource competition in the degradation process is significant. Overall, this well-designed and novel study provides a valuable framework for future research on engineering synthetic gene circuits in mammalian cells. Here are some minor comments:

1. The authors did an excellent job designing the experiment to isolate degradation as the sole factor leading to changes in gene expression. This allowed them to conclude that the changes were due to competition for degradation resources rather than transcriptional or translational resources. However, in Fig. 4D, where a library of promoters is used, both protein and mRNA levels are altered. While I understand that the authors intended to use this for testing the monitoring system, it is essential to clarify whether there is competition in both transcription/translation and degradation processes here. Is there any degradation competition without using specific degradation tags?
2. The font sizes in Figs. 3 and 4 are small for some labels. It is suggested to reorganize the figures to enhance the font size.
3. Line 177 appears to be cut off. Please complete the sentence.
4. Lines 60-61 consist of only two sentences. It might be better to combine them with the next paragraph for better flow.

Reviewer #2

(Remarks to the Author)

Resource competition has become a critical consideration when designing synthetic biology constructs. This work by Gabrielli et al. expands the current understanding of proteolytic resource competition with synthetic systems to mammalian cells and degrons specific to those systems. The findings show coupling of specific degrons and the potential of a few orthogonal systems. The authors also test the orthogonality of two inducible degrons, which are a key design component in some synthetic constructs and have significance to the field. And the authors test the effects of proteolytic resource competition on an integrated monitor, which shows heterologous expression may affect genomic systems. As the authors note, their findings will have future applications in the design of synthetic systems in mammalian cells, while also informing our basic understanding of resource availability in natural systems.

My main critique of the work is that additional controls and analysis are needed to confirm the validity of their findings regarding resources competition between specific degrons.

- When testing the degrons, the authors use eGFP as the control for change in eGFP levels with the different degron tags. However, the authors do not show mKate without a degron tag as a control. An mKate (no tag) monitor would also address Lines 124-132 where the authors “speculate that this restoration is not attributed to increased expression levels of the mKATE gene but rather results from reduced degradation of mKATE”. A mKate monitor is needed as a control to fully understand the significance of the results.

- When demonstrating cross-talk between degrons it is best to show competition with the degrons on either fluorescent proteins. This accounts for any small differences in amino acid sequences or expression levels that may result from differences in plasmid backbones and promoters (Butzin and Mather 2017). The authors do not meet this condition for many degrons. For example, the authors show crosstalk between eGFP-MODCPEST and mKate-PEST in figure 1, but they never show crosstalk between eGFP-PEST and mKate-MODCPEST. By not showing the reverse, the competition may be specific to eGFP-MODCPEST rather than generally extended to the presence of the MODCPEST tag. While the unidirectional findings are of interest, demonstrating the degrons compete for resources from either expression system would solidify the findings.

- In Figure 2, the authors chose not to show the p-values for the eGFP readings 'for clarity'. In the manuscript, it is not clearly explained why the authors are basing the significance of their findings solely on mKate monitor expression levels as further analysis should be completed to capture the scope of the work. The data should have direct comparisons between the different eGFP-degron levels under different monitors. For example, is the eGFP-PEST expression with the mKATE-PEST monitor significantly different from eGFP-PEST expression with an untagged mKATE monitor? I would suggest a complementary, supplemental figure comparing differences in eGFP-degron expression with mKate monitors.

Additional suggestions for revisions:

- Resource competition would generally be expected between proteins with the same degron; however, eGFP-PEST did not affect the levels of mKate-PEST in HEK293T cells. Additionally, the HEK293T results show no impact of any eGFP degron combination on UbVR monitor. Were the expression levels simply too low to measure crosstalk? The authors should incorporate these results when discussing their findings and address further discuss why the results differ between the two cell lines.

- Figure 2: The tagged-eGFP are shown in a different order on the x-axis across the figure. It appears the order is based on mKate fluorescence. I suggest using the same order to simplify finding specific degrons for the readers.

- Lines 150-151: The authors state that "In CHO-K1 cells, C-terminal degrons, PEST, MODCPEST, and UbVR-PEST showed a higher propensity for coupling by causing an increase in the levels of the capacity monitor, irrespective of the capacity monitors tested" However, eGFP-MODCPEST with UbR-mKate did not have significance according to Figure 2B.

- Line 192: authors refer to "previous work" and it is unclear to what previous work they are referring.

- Lines 311-312 point out the advantage of the monitor system as being "live and dynamic". However, the authors do not present any time course results confirming that the system does capture dynamic expression levels. A time course experiment would be of interest to further support the work in the manuscript.

Version 1:

Reviewer comments:

Reviewer #1

(Remarks to the Author)

The authors have addressed my comments satisfactorily. I appreciate the effort they put into conducting additional experiments.

Reviewer #2

(Remarks to the Author)

This work is an excellent addition to the field of synthetic biology as it improves our understanding of host-burden in mammalian cell lines and develops new tools for constructing orthogonal synthetic systems. The results can be generally extended to emphasize the complexity of intracellular interactions between protein production and cellular degradation, which lead to resource competition in mammalian cell systems. The additional experiments and changes made by the authors address my previous concerns. I am happy to recommend the article for publication.

Response to the reviewers

Reviewer #1 (Remarks to the Author):

In this work, the authors investigated whether competition for cellular degradation resources could affect synthetic gene expression levels in a mammalian system. They utilized a library of degradation domains to characterize this competition in both transient and genomically integrated systems. Through systematic measurements across multiple cell lines and rigorous analysis, the authors confirmed that resource competition in the degradation process is significant. Overall, this well-designed and novel study provides a valuable framework for future research on engineering synthetic gene circuits in mammalian cells.

We thank the reviewer for confirming our study is novel, well-designed and valuable for future research work in the field.

1. The authors did an excellent job designing the experiment to isolate degradation as the sole factor leading to changes in gene expression. This allowed them to conclude that the changes were due to competition for degradation resources rather than transcriptional or translational resources. However, in Fig. 4D, where a library of promoters is used, both protein and mRNA levels are altered. While I understand that the authors intended to use this for testing the monitoring system, it is essential to clarify whether there is competition in both transcription/translation and degradation processes here. Is there any degradation competition without using specific degradation tags?

We thank the reviewer for mentioning that we did an excellent job in designing our experiments. We agree with the reviewer on the need to clarify the data presented in Figure 4D and whether there is competition for transcription, translation and degradation or if the absence of degradation tags on the EGFP test constructs, as for the case shown in Figure 4D, implies that only transcription and translation resources are involved in this case. With the aim to answer the reviewer question and to answer a request from Reviewer 2 (reported below) we performed new experiments where multiple EGFP expressing constructs, one untagged and others tagged for degradation, were co-transfected with an untagged mKate monitor. We thought that competition with an untagged monitor would allow us to quantify competition for transcription and translation only, while comparing the decrease of the monitor when a degradation tag is also added would allow quantification of the change in the monitor levels due to the presence of a degradation tag on the competing EGFP constructs. Data are now reported in two new figures (new Figure 1 and Figure S1 pasted here below and added to the manuscript) and confirm that the untagged monitor levels in the case of competition with tagged or untagged EGFP do not change or at most decrease, suggesting that the absence of a tag on one of the competing constructs alleviates competition for degradation resources. We also found evidence in the literature supporting our findings. In Yen et al (Science 2008, DOI: 10.1126/science.1160489) authors report EGFP half-life to be EGFP, $t_{1/2} = 24$ hours. Considering that HEK293T cells have a division time of ~21-22 hours in our growing conditions (previously established in Grob et al. Nature Communications 2024 DOI: 10.1038/s41467-023-44396-4), we speculate that in the absence of a degradation tag, dilution of the EGFP is mainly driven by cell division

and thus dilution, rather than active degradation. Degradation may surely still occur but as a minor contributor to decrease in intracellular EGFP.

Figure 1. Probing degradation-induced coupling via capacity monitors. (A) Simplified diagram of the UPS degradation pathway followed by experimental workflow schematics: A library of test constructs, tagged with degrons, is co-transfected into HEK293T and CHO-K1 cell lines along with a destabilized capacity monitor. Intracellular protein levels are quantified using flow cytometry. The competition for degradation resources can be visualized by plotting the levels of the capacity monitor protein against the levels of the test constructs' protein. In this plot, the constructs located at the bottom left corner represent the most efficient candidates. **(B)** Proof-of-concept experiment where a destabilized monitor, with a PEST degron, is co-transfected with a test construct tagged with the MODCPST degron in HEK293T and CHO-K1 cells. **(C)** Proof-of-concept experiment where an untagged stable monitor is co-transfected with a test construct tagged with the MODCPST degron in HEK293T and CHO-K1 cells. Test plasmid and capacity monitor intracellular protein levels are reported as mean fluorescence (arbitrary units) \pm std. The table reports $1/(\text{mKate of EGFP-tagged condition}/\text{mKate of EGFP untagged condition})$ to quantify change in degradation capacity upon adding the degron \pm the ratio of the standard deviation to the mKate mean multiplied by change in capacity. Data derive from three independent experiments each comprised of two biological repeats. A two-tailed Student T-Test was used, where P values are denoted as follows: ****<math><0.00005</math>, ***<math><0.0005</math>, **<math><0.0005</math>, *<math><0.005</math>, <math><0.05</math>. The number of biological repeats for each sample and exact P values are reported in Source data file.

Figure S1. An untagged mKate monitor results in the absence of competition for degradation. A) Schematics of the genetic construct library where the untagged mKate monitor was co-transfected with seven EGFP-expressing test constructs tagged with UbM, UbR, 2xUbAV, UbVR, PEST or MODCPEST in both HEK293T and CHO-K1. **B)** Flow cytometry data for the untagged monitor in CHO-K1 (top) and HEK293T (bottom). Test plasmid and capacity monitor fluorescence levels are reported as mean fluorescence (arbitrary units) ± std. Data derived from three independent experiments each with two biological repeats. A two-tailed Student T-Test was used, where P values are denoted as follows: *****<0.00005, ****<0.0005, ***<0.0005, **<0.005, *<0.05, and it was annotated only on the mKate charts with reference to the control, EGFP versus capacity monitor, for clarity purposes. Tables report 1/(mKate of EGFP-tagged condition/mKate of EGFP untagged condition) to quantify change in degradation capacity upon adding the degron ± the ratio of the standard deviation to the mKate mean multiplied by change in capacity. The number of biological repeats for each sample and exact P values are reported in Source data file.

2. The font sizes in Figs. 3 and 4 are small for some labels. It is suggested to reorganize the figures to enhance the font size.

We thank the reviewer for their suggestion. Both figures 3 and 4 have now been edited to enhance the font size.

3. Line 177 appears to be cut off. Please complete the sentence.

We thank the reviewer for noticing this. We have now fixed the sentence.

4. Lines 60-61 consist of only two sentences. It might be better to combine them with the next paragraph for better flow.

We thank the reviewer for noticing this. We have now accepted the suggestion.

Reviewer #2 (Remarks to the Author):

Resource competition has become a critical consideration when designing synthetic biology constructs. This work by Gabrielli et al. expands the current understanding of proteolytic resource competition with synthetic systems to mammalian cells and degrons specific to those systems. The findings show coupling of specific degrons and the potential of a few orthogonal systems. The authors also test the orthogonality of two inducible degrons, which are a key design component in some synthetic constructs and have significance to the field. And the authors test the effects of proteolytic resource competition on an integrated monitor, which shows heterologous expression may affect genomic systems. As the authors note, their findings will have future applications in the design of synthetic systems in mammalian cells, while also informing our basic understanding of resource availability in natural systems.

We thank the reviewer for confirming that the findings presented in our manuscript will have future implications in the design of synthetic constructs and that they will inform basic understanding on resource allocation in natural systems.

My main critique of the work is that additional controls and analysis are needed to confirm the validity of their findings regarding resources competition between specific degrons.

- 1) When testing the degrons, the authors use EGFP as the control for change in EGFP levels with the different degon tags. However, the authors do not show mKate without a degon tag as a control. An mKate (no tag) monitor would also address Lines 124-132 where the authors “speculate that this restoration is not attributed to increased expression levels of the mKate gene but rather results from reduced degradation of mKate”. A mKate monitor is needed as a control to fully understand the significance of the results.

We thank the reviewer for the comment that gives us the opportunity to expand the significance of our work. We have now performed new experiments to include an untagged mKate monitor for the experiments in Figure 1 (now replaced with a new Figure 1, shown here below and added to the manuscript).

Figure 1. Probing degradation-induced coupling via capacity monitors. (A) Simplified diagram of the UPS degradation pathway followed by experimental workflow schematics: A library of test constructs, tagged with degrons, is co-transfected into HEK293T and CHO-K1 cell lines along with a destabilized capacity monitor. Intracellular protein levels are quantified using flow cytometry. The competition for degradation resources can be visualized by plotting the levels of the capacity monitor protein against the levels of the test constructs' protein. In this plot, the constructs located at the bottom left corner represent the most efficient candidates. (B) Proof-of-concept experiment where a destabilized monitor, with a PEST degron, is co-transfected with a test construct tagged with the MODCPEST degron in HEK293T and CHO-K1 cells. (C) Proof-of-concept experiment where an untagged stable monitor is co-transfected with a test construct tagged with the MODCPEST degron in HEK293T and CHO-K1 cells. Test plasmid and capacity monitor intracellular protein levels are reported as mean fluorescence (arbitrary units) ± std. The table reports $1/(\text{mKate of EGFP-tagged condition}/\text{mKate of EGFP untagged condition})$ to quantify change in degradation capacity upon adding the degron ± the ratio of the standard deviation to the mKate mean multiplied by change in capacity. Data derive from three independent experiments each comprised of two biological repeats. A two-tailed Student T-Test was used, where P values are denoted as follows: ****<math><0.00005</math>, ****<math><0.0005</math>, ***<math><0.0005</math>, **<math><0.005</math>, *<math><0.05</math>. The number of biological repeats for each sample and exact P values are reported in Source data file.

As shown in the new Figure 1, when the mKate monitor with no degradation tag is co-transfected in HEK293T and in CHO-K1 with an untagged or with a MODPEST-tagged EGFP, the expression levels of mKate do not increase, differently from what observed in the case of competition of the tagged monitor. When the mKate monitor is instead tagged for degradation, competition with MODPEST-tagged EGFP leads to an increase in expression compared to the case of an untagged EGFP.

We also performed the same control for the experiments in Figure 2, where competition of EGFP constructs tagged with different degrons and untagged mKate monitor is shown (now new Figure S1, shown here below and added to the manuscript).

Figure S1. An untagged mKate monitor results in the absence of competition for degradation. A) Schematics of the genetic construct library where the untagged mKate monitor was co-transfected with seven EGFP-expressing test constructs tagged with UbM, Ubr, 2xUbAV, UbVR, PEST or MODCPEST in both HEK293T and CHO-K1. **B)** Flow cytometry data for the untagged monitor in CHO-K1 (top) and HEK293T (bottom). Test plasmid and capacity monitor fluorescence levels are reported as mean fluorescence (arbitrary units) ± std. Data derived from three independent experiments each with two biological repeats. A two-tailed Student T-Test was used, where P values are denoted as follows: ****<math><0.00005</math>, ****<math><0.0005</math>, ***<math><0.0005</math>, **<math><0.005</math>, *<math><0.05</math>, and it was annotated only on the mKate charts with reference to the control, EGFP versus capacity monitor, for clarity purposes. Tables report $1/(\text{mKate of EGFP-tagged condition}/\text{mKate of EGFP untagged condition})$ to quantify change in degradation capacity upon adding the degron ± the ratio of the standard deviation to the mKate mean multiplied by change in capacity. The number of biological repeats for each sample and exact P values are reported in Source data file.

Again, we confirmed that untagged mKate monitor levels do not increase above the levels displayed by mKate in competition with an untagged EGFP with statistical significance for any of the combinations. We speculate that the increase in monitor expression observed when the monitor is tagged for degradation (Figure 2) is thus due to competition for degradation. Interestingly, we observed that for some of the degrons, untagged mKate expression decreases below the levels observed when mKate competes with untagged EGFP, suggesting that, for some combinations, the presence of the degron leads to increased competition for gene expression upstream of degradation.

In response to the reviewer in Question 2 below, we also performed new experiments to consider the case of EGFP untagged monitor in competition with tagged mKate constructs. This complemented the request from Reviewer 2 to perform experiments where tagged mKate constructs compete with an EGFP monitor. These are reported in the new Figure S4 in the manuscript, reported here below:

Figure S4. An untagged EGFP monitor results in the absence of competition for degradation. A) Schematics of the genetic construct library where the untagged EGFP monitor was co-transfected with six mKate-expressing test constructs tagged with UbM, UbR, UbVR, PEST or MODCPEST in both HEK293T and CHO-K1. **B)** Flow cytometry data for the untagged monitor in CHO-K1 (top) and HEK293T (bottom). 2xUbAV-mKate is missing due to a repeated failure

in DNA assembly of the construct. Test plasmid and capacity monitor fluorescence levels are reported as mean fluorescence (arbitrary units) \pm std. Data derived from three independent experiments each with two biological repeats. A two-tailed Student T-Test was used, where P values are denoted as follows: ***** <0.00005 , **** <0.0005 , *** <0.0005 , ** <0.005 , * <0.05 , and it was annotated only on the mKate charts with reference to the control, EGFP versus capacity monitor, for clarity purposes. Tables report $1/(\text{EGFP of mKate-tagged condition}/\text{EGFP of mKate untagged condition})$ to quantify change in degradation capacity upon adding the degron \pm the ratio of the standard deviation to the EGFP mean multiplied by change in capacity. The number of biological repeats for each sample and exact P values are reported in Source data file.

Again, we did not observe any statistically significant increase in the monitor level, as previously confirmed for the untagged mKate control. Notably, for PEST, MODCPEST and UbVR-PEST in CHO-K1 an increase in the level of mKate construct expression was observed.

Overall, our results confirmed that for untagged mKate and EGFP no significant increase in monitor levels occur as a consequence of competition. Interestingly, we also observed that different degrons led to more or less pronounced decrease of mKate/EGFP levels, suggesting that the presence of the degron impacts gene expression competition upstream of degradation possibly due to impact in transcription and translation efficiency.

We have now incorporated this new information into the main manuscript text adding the following paragraphs highlighted in red in the text:

“and performed a parallel control experiment where the mKate monitor is instead not tagged for degradation”

“Such increase was not observed in the absence of PEST on the mKate coding sequence (Figure 1C), suggesting the observed restoration was not attributed to increased expression levels of the mKate gene but rather a result of reduced degradation of mKate, a phenomenon akin to previously documented evidence in bacterial systems”

“As a control, the same experiment was performed in parallel with EGFP constructs tagged with the different degrons co-transfected with untagged mKate (Figure S1). Again, we confirmed that untagged mKate monitor levels do not increase above the levels displayed by mKate in competition with an untagged EGFP with statistical significance for any of the combinations. We speculate that the increase in monitor expression observed when the monitor is tagged for degradation (Figure 2) is due to competition for degradation. Interestingly, we observed that for some of the degrons, untagged mKate expression decreases below the levels observed when mKate competes with untagged EGFP, suggesting that, for some combinations, the presence of the degron leads to increased competition for gene expression upstream of degradation possibly due to impact in transcription and translation efficiency”

“In parallel, we also included control competition experiments between the same tagged EGFP/mKate constructs and untagged mKate/EGFP monitors to confirm that the increase in construct expression levels are indeed due only to competition for degradation. Our results confirmed that for untagged mKate and EGFP no significant increase in monitor levels occur as a consequence of competition. Interestingly, we also observed that different degrons led to more or less pronounced decrease of mKate/EGFP levels, suggesting that the presence of the degron impacts gene expression competition upstream of degradation.

This highlights a more complex scenario than previously hypothesised and adds to the importance to characterise the impact of context on the specific system considered”

- 2) When demonstrating crosstalk between degrons it is best to show competition with the degrons on either fluorescent protein. This accounts for any small differences in amino acid sequences or expression levels that may result from differences in plasmid backbones and promoters (Butzin and Mather 2017). The authors do not meet this condition for many degrons. For example, the authors show crosstalk between EGFP-MODCPEST and mKate-PEST in figure 1, but they never show crosstalk between EGFP-PEST and mKate-MODCPEST. By not showing the reverse, the competition may be specific to EGFP-MODCPEST rather than generally extended to the presence of the MODCPEST tag. While the unidirectional findings are of interest, demonstrating the degrons compete for resources from either expression system would solidify the findings.

We thank the reviewer for the suggested experiments as this allows us to add to our initial tests. We assembled a new library of constructs where mKate is tagged with the same degrons initially adopted in Figure 2 to tag the EGFP constructs (to be noted that only six are present for this library as the 2xUbAV combination did not successfully clone-this is specified in the manuscript). This new library was used to perform new competition experiments in HEK293T and CHO-K1 between the tagged mKate library and three different EGFP tagged monitors, similar and opposite to Figure 2, as suggested by the reviewer. These experiments were aimed at characterising if the initially observed competition patterns were degrons specific or if the combination of the degron and the cognate protein sequence impacts the pattern of competition. By looking at the new results (now displayed in a new Figure S2, for HEK293T, and Figure S3, for CHO-K1, reported below and added to the manuscript) we observed that the competition pattern is indeed impacted by the combination of degron and protein considered. Specifically, we observed that in the case of EGFP monitors in HEK293T, competition was only detected with UbR and UbM N-terminal degrons while in CHO-K1 no significant increase in monitor levels was detected. Therefore, protein context, cell line, and we speculate, position of the degron on the protein sequence, appear to play a major role in the establishment of protein degradation bottlenecks.

Figure S2. Characterisation of a library of degrons with three co-transfected EGFP monitors in HEK293T cells. **A)** Schematics of the genetic construct library considered where three EGFP-expressing monitors tagged with UbR, UbVR and PEST respectively were co-transfected with six mKate-expressing test constructs tagged with UbM, UbR, UbVR, PEST or MODCPEST. **B)** Flow cytometry data for the EGFP monitor with PEST. **C)** Flow cytometry data for the EGFP monitor with UbR. **D)** Flow cytometry data for the EGFP monitor with UbVR. 2xUbAV-mKate is missing due to a repeated failure in DNA assembly of the construct. Test plasmid and capacity monitor fluorescence levels are reported as mean fluorescence (arbitrary units) ± std. Data derived from three independent experiments each with two biological repeats. A two-tailed Student T-Test was used, where P values are denoted as follows: ****<0.00005, ***<0.0005, **<0.0005, *<0.005, *<0.05, and it was annotated only on the mKate charts with reference to the control, EGFP versus capacity monitor, for clarity purposes. Tables report 1/(EGFP of mKate-tagged condition/EGFP of mKate untagged condition) to quantify change in degradation capacity upon adding the degron ± the ratio of the standard deviation to the EGFP mean multiplied by change in capacity. The number of biological repeats for each sample and exact P values are reported in Source data file.

Figure S3. Characterisation of a library of degrons with three co-transfected EGFP monitors in CHO-K1 cells. A) Schematics of the genetic construct library considered where three EGFP-expressing monitors tagged with UbR, UbM, UbVR and PEST respectively were co-transfected with six mKate-expressing test constructs tagged with UbM, UbR, UbVR, PEST or MODCPEST. **B)** Flow cytometry data for the EGFP monitor with PEST. **C)** Flow cytometry data for the EGFP monitor with UbR. **D)** Flow cytometry data for the EGFP monitor with UbVR. 2xUbAV-mKate is missing due to a repeated failure in DNA assembly of the construct. Test plasmid and capacity monitor fluorescence levels are reported as mean fluorescence (arbitrary units) \pm std. Data derived from three independent experiments each with two biological repeats. A two-tailed Student T-Test was used, where P values are denoted as follows: ***** <0.00005 , **** <0.0005 , *** <0.0005 , ** <0.005 , * <0.05 , and it was annotated only on the mKate charts with reference to the control, EGFP versus capacity monitor, for clarity purposes. Tables report $1/(\text{EGFP of mKate-tagged condition}/\text{EGFP of mKate untagged condition})$ to quantify change in degradation capacity upon adding the degron \pm the ratio of the standard deviation to the EGFP mean multiplied by change in capacity. The number of biological repeats for each sample and exact P values are reported in Source data file.

We have now included this information in the main text by adding the following paragraphs in the manuscript:

“To capture not only the impact of the cell host but also the one of the protein context on competition, we then assembled a new library of constructs where mKate bears the same degrons previously used for EGFP and EGFP now functions as the capacity monitor. Experiments were performed in both HEK293T (Figure S2) and CHO-K1 cells (Figure S3). For HEK293T cells, competition was only detected with UbR and UbM N-terminal degrons while in CHO-K1 cells no significant increase in monitor levels was detected. We also performed experiments with their respective untagged EGFP monitor controls, similar to what previously done for the mKate monitor (Figure S4). Again, we did not observe any statistically significant increase in the monitor level, as previously confirmed for the untagged mKate control. Notably, for PEST, MODCPEST and UbVR-PEST in CHO-K1 an increase in the level of mKate construct expression was observed (Figure S4), potentially providing an explanation for the competition pattern observed for the EGFP-monitors in Figure S3. Overall, from our data, protein context, cell host, and, potentially, position of the degron on the protein sequence, appear to play a major role in shaping competition for protein degradation”

And in the discussion:

“In order to capture the effect of the cellular context on competition for degradation, we first considered the case of an mKate monitor bearing one of three degradation tags (PEST, UbR or UbvR) in competition with EGFP constructs bearing one of seven degradation tags (UbVR, 2xUbAV, UbR, UbM, MODCPEST, PEST and a composite degron with an N-terminal UbVR and a C-terminal PEST) in two different cell lines, HEK293T and CHO-K1. Our data highlight that competition for degradation is cell line dependent. Specifically, in HEK293T, the UbVR-mKate monitor displayed no significant competition with any of the EGFP tagged constructs while the PEST-mKate monitor did not display competition with the corresponding PEST-tagged EGFP construct. This was not the case for experiments performed in CHO-K1 where we instead observed competition for the same combinations resulting in an increase in monitor expression levels. While we do not have a justification for the observed difference, we speculate that, as previously observed for competition imposed by other genetic elements like promoters and PolyA signals³⁵, different cellular environments and differences in basal levels of resources related to gene expression regulation could lead to variations in competition profiles. Additionally, minor cell line specificity when comparing degron activity was also previously reported with multiple degrons including UbM and UbR also featured in our study, in HEK293, HeLa, hMSCtert and CHO cells¹⁷. Finally, the observed low levels of expression resulting from enhanced degradation could make it more challenging to detect a statistically significant resource competition effect given the intrinsic expression variability which in our data appears higher in HEK293T than in CHO-K1”

“To account for protein context effects on competition, we then performed experiments for the case where the monitor is EGFP instead of mKate, adopting the same combination of degron tags. For the case of the mKate capacity monitors, C-terminal degrons such as PEST and MODCPEST showed a higher degree of coupling across cell lines, while when an EGFP capacity monitor is adopted, competition was only detected with UbR and UbM N-terminal degrons in HEK293T for two of the three monitors considered. Interestingly,

in these experiments the UbVR-EGFP monitor displayed competition when co-transfected with the mKate library in HEK293T while the UbR-EGFP monitor did not yield significant competition, as previously observed for the UbR-mKate monitor. Previous studies reported how the degradation strength of a given degron is impacted by the protein associated with it, leading to significantly different degradation strength for the same degrons¹⁵. Degron-mediated degradation efficiency has also been reported to be dictated by the presence of specific structural and conformational elements. Missense mutations in human DHFR were found to cause conformational changes that led the wildtype DHFR to expose an otherwise hidden degron, with consequent increase in DHFR degradation⁵¹. Protein domains termed “degronons”, while not directly recruiting the E3 ligase, were found to coordinate 26S proteasome binding its substrate. Therefore, changing the protein but maintaining the degron could weaken or strengthen its activity based on the presence or absence of degronons on the new protein”.

“Our data highlight how context dependencies, specifically the protein and cell host context together with position of the degron on the tagged protein, play a crucial role in shaping competition profiles and the activity of multiple mammalian degrons. They also identify designs with improved orthogonality in resource usage, making them amenable to synthetic systems with multiple degron-tagged proteins”.

- 3) In Figure 2, the authors chose not to show the p-values for the EGFP readings ‘for clarity’. In the manuscript, it is not clearly explained why the authors are basing the significance of their findings solely on mKate monitor expression levels as further analysis should be completed to capture the scope of the work. The data should have direct comparisons between the different EGFP-degron levels under different monitors. For example, is the EGFP-PEST expression with the mKate-PEST monitor significantly different from EGFP-PEST expression with an untagged mKate monitor? I would suggest a complementary, supplemental figure comparing differences in EGFP-degron expression with mKate monitors.

We thank the reviewer. As requested, we have now added statistics to highlight differences in EGFP monitors for the newly performed experiments in the new Figure S2 and S3.

Additional suggestions for revisions:

Resource competition would generally be expected between proteins with the same degron; however, EGFP-PEST did not affect the levels of mKate-PEST in HEK293T cells. Additionally, the HEK293T results show no impact of any EGFP degron combination on UbVR monitor. Were the expression levels simply too low to measure crosstalk? The authors should incorporate these results when discussing their findings and address further discuss why the results differ between the two cell lines.

We thank the reviewer for the comment. We have now considered the requested comments by adding to the discussion the following paragraph:

“In order to capture the effect of the cellular context on competition for degradation, we first considered the case of an mKate monitor bearing one of three degradation tags (PEST, UbR or UbvR) in competition with EGFP constructs bearing one of seven degradation tags (UbVR, 2xUbAV, UbR, UbM, MODCPEST, PEST and a composite degron with an N-terminal UbVR and a C-terminal PEST) in two different cell lines,

HEK293T and CHO-K1. Our data highlight that competition for degradation is cell line dependent. Specifically, in HEK293T, the UbVR-mKate monitor displayed no significant competition with any of the EGFP tagged constructs while the PEST-mKate monitor did not display competition with the corresponding PEST-tagged EGFP construct. This was not the case for experiments performed in CHO-K1 where we instead observed competition for the same combinations resulting in an increase in monitor expression levels. While we do not have a justification for the observed difference, we speculate that, as previously observed for competition imposed by other genetic elements like promoters and PolyA signals³⁵, different cellular environments and differences in basal levels of resources related to gene expression regulation could lead to variations in competition profiles. Additionally, minor cell line specificity when comparing degron activity was also previously reported with multiple degrons including UbM and UbR also featured in our study, in HEK293, HeLa, hMSCtert and CHO cells¹⁷. Finally, the observed low levels of expression resulting from enhanced degradation could make it more challenging to detect a statistically significant resource competition effect given the intrinsic expression variability which in our data appears higher in HEK293T than in CHO-K1".

- Figure 2: The tagged-EGFP are shown in a different order on the x-axis across the figure. It appears the order is based on mKate fluorescence. I suggest using the same order to simplify finding specific degrons for the readers.

As per the reviewer`s suggestion, the order has now been fixed across all manuscript figures.

- Lines 150-151: The authors state that "In CHO-K1 cells, C-terminal degrons, PEST, MODCPEST, and UbVR-PEST showed a higher propensity for coupling by causing an increase in the levels of the capacity monitor, irrespective of the capacity monitors tested" However, EGFP-MODCPEST with UbR-mKate did not have significance according to Figure 2B.

We thank the reviewer for noticing this. We have now edited the sentence replacing it with the following: "In CHO-K1 cells, the C-terminal degrons PEST and UbVR-PEST showed a higher propensity for coupling"

- Line 192: authors refer to "previous work" and it is unclear to what previous work they are referring.

We thank the reviewer for noticing the omission. We have now added the appropriate reference that was missing by mistake.

Lines 311-312 point out the advantage of the monitor system as being "live and dynamic". However, the authors do not present any time course results confirming that the system does capture dynamic expression levels. A time course experiment would be of interest to further support the work in the manuscript.

We agree with the reviewer that the mention to “live and dynamic” tracking is misleading and should not be present in the text as we do not show dynamic measurements in our manuscript. We have not removed these words from the text.